# Clinical Characteristics and Follow-Up of Children with Primary Haematogenous Osteomyelitis and Septic Arthritis: Eight Years of Experience from Hungary

**DOI:** 10.3390/antibiotics14080821

**Published:** 2025-08-11

**Authors:** Szofia Hajósi-Kalcakosz, Erzsébet Varga, Dorottya Őri, Csaba Ráskai, Borbála Zsigmond, Beáta Visy, Ferenc Fekete, Andrea Horváth, Orsolya Dobay, Bálint Gergely Szabó

**Affiliations:** 1Department of Infectious Diseases, Heim Pál National Paediatric Institute, Üllői út 86, H-1089 Budapest, Hungary; szofia.kalcakosz@heimpalkorhaz.hu (S.H.-K.);; 2Tropical Medicine Specialist Training, Faculty of Medicine, Semmelweis University, Üllői út 26, H-1085 Budapest, Hungary; 3Department of Psychiatry, Heim Pál National Paediatric Institute, Üllői út 86, H-1089 Budapest, Hungary; 4Institute of Behavioural Sciences, Semmelweis University, Nagyvárad tér 4, H-1089 Budapest, Hungary; 5Department of Orthopaedic Surgery, Heim Pál National Paediatric Institute, Üllői út 86, H-1089 Budapest, Hungary; 6Department of Paediatric Infectious Diseases, St. George’s University Hospitals NHS Foundation Trust, Blackshaw Rd, London SW17 0QT, UK; 7Institute of Medical Microbiology, Semmelweis University, Nagyvárad tér 4, H-1089 Budapest, Hungary; 8Division of Infectious Diseases, Department of Internal Medicine and Haematology, Semmelweis University, H-1097 Budapest, Hungary; 9Doctoral College, Semmelweis University, Üllői út 26, H-1085 Budapest, Hungary; 10South Pest Central Hospital, National Institute of Haematology and Infectology, Albert Flórián út 5-7, H-1097 Budapest, Hungary

**Keywords:** acute haematogenous osteomyelitis, bone and joint infection, paediatric, septic arthritis, *Staphylococcus aureus*

## Abstract

**Introduction:** Paediatric acute haematogenous bone and joint infections (BJIs) are serious conditions. This study aimed to analyse the characteristics of paediatric acute haematogenous osteomyelitis (AHO) and septic arthritis (SA) in Hungary, with a focus on causative pathogens, clinical outcomes, and long-term complications. **Methods**: A retrospective cohort study was conducted at a Hungarian tertiary referral centre between 2015 and 2022. Children aged 18 years or younger diagnosed with acute haematogenous osteomyelitis (AHO) or septic arthritis (SA) within two months of symptom onset were included. Exclusion criteria were chronic infection, post-operative infections, or wound-related infections. Complicated AHO was defined by intraosseous abscess or necrosis confirmed radiologically or intraoperatively. The primary outcome was surgical intervention beyond 30 days after diagnosis; secondary outcomes included long-term complications. **Results**: Forty patients were included (77.5% male, median age 8.7 years). AHO was diagnosed in 8 patients (20.0%), complicated AHO in 22 (55.0%), and SA in 10 (25.0%). MRI had the highest diagnostic sensitivity (97.0%). Pathogens were identified in 72.5% of cases; *Staphylococcus aureus* (*S. aureus*) was most common (57.5%), followed by *Salmonella* and *Streptococcus pyogenes* (5% each). Surgery was required in 90.0% of SA cases, 77.2% of complicated AHO, and 37.5% of uncomplicated AHO. Long-term complications occurred in 10%, mainly with *S. aureus* and complicated AHO. **Conclusions**: Paediatric BJIs, especially due to *S. aureus*, often require surgery and cause long-term sequelae.

## 1. Introduction

Acute haematogenous bone and joint infections (BJIs) in children can manifest as primary haematogenous osteomyelitis (AHO), septic arthritis (SA), or a combination of both. These conditions arise when microorganisms, most commonly *Staphylococcus aureus*, spread through the bloodstream to infect the bone, bone marrow, or synovial joints, sometimes leading to rapid tissue destruction. Other significant pathogens include *Streptococcus* species, *Kingella kingae,* and *Escherichia coli*, among others [1]. The infections often present with localized pain, swelling, warmth, and systemic signs such as fever, although symptoms may be subtle. Diagnosis of these infections requires a combination of clinical assessment, laboratory, microbiological tests, and imaging studies. Magnetic resonance imaging (MRI) is considered the gold standard for early detection due to its high sensitivity, while ultrasonography aids in identifying joint effusions in cases of SA (Figure 1) [2,3]. Microbiological cultures can be negative in a significant proportion of cases; therefore, molecular methods such as the polymerase chain reaction (PCR) of clinical samples are increasingly used to detect difficult-to-culture pathogens.

AHO and SA carry significant risks of serious complications if treatment is delayed or ineffective. These can include chronic osteomyelitis, avascular necrosis, growth disturbances, pathologic fractures, and in severe cases, life-threatening sepsis. Treatment formerly involved a prolonged course of intravenous antibiotics, followed by oral antibiotics tailored to the causative pathogen, alongside surgical interventions when necessary. Recently, shorter intravenous therapies have been recommended, and there are now randomised-controlled trials of oral-only treatment options [4]. Close follow-up over at least one year is recommended to monitor and manage potential long-term sequelae [1]. From a public health perspective, bone and joint infections in children represent a persistent burden, both globally and regionally. These infections lead to prolonged hospital stays, increased healthcare costs, and long-term disability, affecting patients’ quality of life and functionality. In Europe, including Hungary, *S. aureus* remains the predominant pathogen, with emerging recognition of *K. kingae* as an important cause in younger children [5].

In recent years, advancements in the management of paediatric BJI have emerged, largely driven by updated clinical guidelines. However, to the best of our knowledge, there are currently no data available on paediatric AHO and SA in Hungary. Therefore, our objective was to evaluate clinical and microbiological data from a high-influx hospital to better understand these rare childhood infections.

## 2. Results

### 2.1. Baseline Characteristics

Baseline characteristics are presented in Table 1. Forty children were enrolled, with a predominance of males (77.5%) and a median age of 8.7 ± 9.9 years. Among them, 75.0% were diagnosed with AHO, while 25.0% were diagnosed with SA. Within the AHO group, the majority of patients were classified as complicated (73.3%), and one patient had spondylodiscitis. Among children aged ≥ 5 years, AHO was more prevalent than SA was (57.5% vs. 12.5%, respectively; *p* = 0.13). Nearly all patients presented with pain and fever at diagnosis. The most frequently affected sites for AHO were the tibia (45.5%), femur (40.9%), humerus (36.7%), and hip joint for the SA (50.0%). In the laboratory results on admission, no significant differences were detected between the AHO and SA groups. 10.0% of children had an initial serum procalcitonin (PCT) concentration < 0.5 ng/mL. The following significant differences were observed in baseline characteristics: patients with AHO had a significantly longer duration of symptoms prior to admission compared to those with SA (5.0 ± 5.0 vs. 2.0 ± 2.0 days, *p* = 0.01). Among presenting symptoms, pseudoparalysis was significantly more common in SA than in AHO (60% vs. 13.3%, *p* = 0.007).

### 2.2. Microbiological Characteristics

The microbiological characteristics are detailed in Table 2 and Figure 2. Altogether, 38/40 (95.0%) blood cultures and 29/40 (72.5%) intraoperative bone specimen cultures were performed in the total cohort, and of these, 24/38 (63.2%) and 24/29 (82.7%) were positive, respectively. Among all children, a causative microorganism was identified in 72.5%, with a higher rate in SA cases (90%) than in complicated AHO (71.4%) and uncomplicated AHO (62.5%) cases. The most common causative pathogen was *Staphylococcus aureus (S. aureus*) (70.8%), followed by *Salmonella* species (spp.) and *Streptococcus pyogenes* (4.2% each). No polymicrobial infections were detected. For the majority of relevant antibiotics, 100% in vitro susceptibility was detected. In vitro antibiotic susceptibility testing revealed moderately reduced susceptibility of *S. aureus* to erythromycin (86.9%), clindamycin (86.9%), and tetracycline (95.6%). A single case of MRSA was identified, corresponding to an incidence rate of 4.3%. Only the sensitivity of *Salmonella* spp. to ciprofloxacin was markedly low. Contaminant organisms were found in three cases (*Bacillus cereus, Micrococcus luteus, Corynebacterium coyleae*). The contamination rates for blood and intraoperative bone cultures were 7.9% and 0%, respectively. For AHO and SA, 66.7% and 70% of the blood cultures and 79.3% and 55.6% of the intraoperative samples were positive, respectively. Among the 11 children with culture-negative results, one had SA, three had uncomplicated AHO, and seven had complicated AHO. All four cases with proven persistent bacteraemia were caused by methicillin-sensitive *Staphylococcus aureus* (MSSA).

### 2.3. Imaging Characteristics

Imaging studies are summarized in Table 3. The overall positivity rates for X-ray radiography, ultrasonography, and magnetic resonance imaging (MRI) were 30%, 85.7%, and 96.9%, respectively. In cases of AHO, half of the positive ultrasound examinations showed soft tissue abnormalities (11/22, 50%). A significant discrepancy was observed in the utilisation of MRI examinations, with a higher proportion of these scans being performed in cases of AHO compared to SA (28/30, 93.3% vs. 4/10, 40%, *p* = 0.001). Only one negative MRI result was obtained on the second day of symptom onset. In cases of AHO, MRI identified intraosseous necrosis, deep vein thrombosis, and bone sequestration in only a minority of patients, with one case reported for each finding. An interim MRI was conducted in 75.0% of cases of uncomplicated AHO and 50.0% of cases of complicated AHO. Significant differences between the AHO and SA groups were found only in the results of the MRI scans. Specifically, the presence of soft tissue abnormalities was found to be significantly more prevalent in AHO than in SA. A total of four patients (12.5%) achieved complete radiological remission on the last MRI.

### 2.4. Antibiotic Treatment Strategies

The characteristics of antimicrobial therapy are reported in Figure 3. All the children received empirical intravenous antibiotics. The most frequently administered intravenous antibiotics were flucloxacillin (55.0%), co-amoxiclav (25.0%), and ceftriaxone (15.0%), whereas first-generation (cephalexin) and second-generation (cefaclor, cefprozil) cephalosporins (35.0%) and co-amoxiclav (35.0%) were the most common oral choices. At the time of switching to oral antibiotics, all the children were afebrile for ≥3 days, with an average serum CRP reduction of 20.42 ± 18.5% from the time of diagnosis. Targeted antibiotic de-escalation was achievable in 18/29 (62.1%), while antibiotic escalation was not required among patients. The durations of parenteral, oral, and total antibiotic therapies did not significantly differ between subcohorts. The median length of stay was not significantly longer for patients with complicated AHO than for those with uncomplicated AHO (20.8 ± 17.2 days vs. 12.3 ± 5.8 days, *p* = 0.28).

### 2.5. Clinical Outcomes and Long-Term Complications

On day +30 post-diagnosis, 9/10 (90.0%) patients with SA, 3/8 (37.5%) patients with uncomplicated AHO, and 17/22 (77.2%) patients with complicated AHO required surgical intervention (*p* = 0.04). Primary surgery was conducted on the fifth hospitalization day on average, with procedures including arthrotomy/arthroscopy for SA and surgical debridement for AHO. Reoperations occurred in six patients (20.1%), five of whom had complicated AHO and one with uncomplicated AHO (*p* = 1.0). The causative microorganism in four (66.7%) of these patients was MSSA. The average time to first reoperation was 15 ± 8 days from hospital admission.

The median follow-up time for all patients was 115 ± 436 days from hospital admission. The overall long-term complication rate was 10.0% (4/40) for the cohort, with stratified rates of 18.2% (4/22), 0%, and 0% for complicated AHO, SA, and uncomplicated AHO, respectively. There was no significant difference in the duration of follow-up between subcohorts (*p* = 0.56). The long-term complications included one pathological fracture, one case of abnormal femur growth, and two cases of permanent hip joint motion reduction. In one patient, total hip replacement was necessary.

## 3. Discussion

This study is the first to analyse the clinical characteristics and treatment of paediatric BJI in Hungary. *S. aureus* was the most common causative pathogen. Infections are mostly managed with prolonged intravenous antimicrobial treatment, and a high proportion of children require surgical intervention, mostly for complicated AHO. A total of 66.7% of all reoperations, cases with persistent bacteraemia, and all long-term complications were attributable to MSSA.

AHO and SA represent a substantial burden on human and public health in Europe. These are serious diseases that require immediate diagnosis and treatment [6]. In high-income countries, the incidence of BJIs is 3–22 per 100,000 per year, with an increasing incidence in recent years [7]. In Lisbon, there was a mean annual incidence of 11.4 cases per 100,000 children, with a 1.8-fold increase observed in the last 5 years of the study period. This increase may be explained by growing awareness, prospective enrolment in studies, and improvements in imaging [8]. In contrast, the estimated incidences of AHO and SA are 2–13 and 4 cases per 10,000 children per year in developed countries, respectively [9]. AHO is 1.2–3.7 times more common in boys than in girls, a ratio that was also observed in our cohort [10]. The vascular supply of growing bones renders children more susceptible to BJI during transient and subclinical bacteraemia [1]. In children, AHO commonly affects the metaphysis of long bones, and spondylodiscitis is observed in only 1–2% of cases [10]. The SA is typically monoarticular, with the knee or hip joint being common sites of involvement [11]. In our study, the most common regions were the tibia and the hip joint.

The pathogen distribution of BJI is dependent on the child’s age. Neonates are vulnerable to *S. aureus*, *Escherichia coli*, *Streptococcus agalactiae (S. agalactiae)*, *Neisseria gonorrhoeae*, *Candida albicans*, and, up to six months of age, *S. aureus*, which is one of the most common agents of osteoarticular infections. In the present study, two children were under three months of age. One of these children was confirmed to have an *S. agalactiae* infection, whereas the other had no detectable pathogen. The sole case of *Serratia marcescens* was below the age of one. Nonetheless, according to recent data with the widespread use of polymerase chain reaction (PCR) among culture-negative cases, *Kingella kingae* was found to be the most common aetiology in some countries of children aged 6–48 months [12]. *K. kingae* infections are typically characterized by a milder presentation, a minor increase in serum inflammatory marker levels, and a more favourable outcome than those caused by other bacteria [13]. *S. aureus* was the most common pathogen in our cohort, with 22 cases of MSSA and one case of methicillin-resistant *Staphylococcus aureus* (MRSA). According to the literature, children with *S. aureus* infection tend to be older, present with fever, and demonstrate a marked acute phase reaction [14]. There is a strong association between MRSA-BJI and purulent complications, an increased likelihood of requiring secondary surgery, and an elevated admission rate to intensive care [15]. In our cohort, 66.7% of the patients underwent reoperation, and all long-term sequelae were attributable to MSSA. Finally, in children younger than 1 year with BJI or in cases with unusual pathogens, testing for primary immunodeficiency is advised [16]. For microbiological testing, samples of blood and surgical samples are subjected to culture. In recent years, synovial fluid PCR in SA has become a common practice because of the rapid identification of difficult-to-culture pathogens. At our institution, PCR is not currently integrated into routine diagnostics, and in 27.5% of our patients, no pathogen was identified by culturing. However, empirical antibiotic therapy was not modified in a greater proportion of culture-negative cases than in culture-positive children, indicating that PCR is probably not universally essential in patient management.

MRI is the most informative imaging method for AHO and is often positive within 3–5 days. In accordance with guidelines, MRI during antibiotic therapy may be performed 2–3 days after diagnosis, provided that the child is in good overall condition [1]. In our cohort, MRI was conducted on 80% of the children one day post-admission. In SA, ultrasonography is the primary imaging modality. Patients with SA are likely to have concomitant AHO when a preoperative MRI is performed; consequently, arthrotomy with bone debridement is performed during surgery [2]. A control MRI scan at the end of therapy is only advised for complicated AHO and is not necessary for uncomplicated AHO and SA [2,3]. In our cohort, 75% of patients with uncomplicated AHO underwent MRI. We do not intend to continue this practice in the future, as the Infectious Diseases Society of America guideline states that follow-up MRI is not always necessary in uncomplicated cases [3].

A timely diagnosis, with the administration of an appropriate antibiotic regimen, represents a key aspect of BJI management. In cases of AHO, blood culture positivity is expected to be observed in approximately one-third of cases, whereas surgical specimens typically yield a positive result 24–48 h after antibiotic therapy initiation [3]. For patients who are clinically stable, a period of 48–72 h is recommended before initiating antibiotic therapy, provided that surgical intervention is anticipated to occur within this timeframe. Nonetheless, pathogen detection remains challenging, with no causative organism identified in over one-third (35%) of patients in some European cohorts [6]. In our cohort, blood culture positivity was 63.2%, which is approximately twice the rate reported in the literature. Furthermore, 82.7% of the intraoperative samples were positive, although 93.1% of the children had already received antibiotics. For empirical antibiotic therapy, the coverage of MSSA is imperative owing to its frequent pathogenicity, for which penicillinase-stable penicillin or first-generation cephalosporins are recommended. Antibiotic administration practices vary significantly across European countries; for example, the UK and Netherlands show higher use of flucloxacillin and ceftriaxone, while Austria and Germany favour clindamycin and cefuroxime [6]. In Hungary, the National Bacteriological Surveillance Report from the National Centre for Public Health reported an MRSA prevalence rate of 11.1–21.5% among all *S. aureus* infections in 2023 [17]. Despite this, we do not have national data on MRSA prevalence among paediatric populations; therefore, it cannot be considered standard to cover MRSA in community-acquired BJI infections. The incidence rate of MRSA infections in our study was low (4.3%). It is hypothesised that this discrepancy is age group-specific in comparison to the national population; therefore, it is conceivable that Hungary may exhibit a lower overall prevalence of MRSA among children. It is also important to note that the above national figure refers to both the community-associated and healthcare-associated infections alike, while there were no hospital-acquired infections among our patients. In view of the low prevalence of MRSA in children, we do not consider empirical MRSA coverage necessary.

The European guidelines have resulted in a reduction in the length of therapy and an earlier switch to oral antibiotics [1]. Currently, a total of 2–3 weeks of antibiotic therapy for SA and 3–4 weeks for AHO are recommended. In the presence of MRSA or complications, in infancy or in cases of pelvic/spinal involvement, a longer course of therapy may be needed. In our study, all the children were hospitalized and initially received intravenous antibiotics. Unfortunately, our median duration of antibiotic therapy exceeded the recommended period, and there was no improvement in this trend after 2017. In addition, a number of articles have been published on reducing the duration of intravenous therapy among children to a few days, after which the treatment is switched to oral medication. Peltola et al. conducted a randomised controlled trial evaluating culture-positive AHO in children [18]. Patients were randomly assigned to receive either clindamycin or a first-generation cephalosporin for 20–30 days, including an intravenous phase of 2–4 days. The authors concluded that the majority of AHO patients should be treated for a period of 20 days, with a short intravenous phase. To date, one single case series, a retrospective case–control study, a nationwide multicentre prospective study, and a randomised-controlled study have demonstrated the efficacy of exclusively oral therapy among children as an alternative [4,19,20,21]. In a Spanish nationwide multicentric registry study, Alcobendas et al. developed criteria for children with BJI for whom an oral-only approach would be beneficial [21]. The criteria for the oral-only approach were 0.5–3 years of age, good overall condition, oral tolerance, no underlying disease, injury, skin infection, recent surgery, complications, CRP ≤ 80 mg/L, ESR/CRP ratio ≥ 0.67, daily follow-up, and informed consent. The first randomised controlled noninferiority trial on oral-only antibiotics has recently been published: 101 patients were randomised to oral treatment, with a primary outcome of sequelae at 6 months [4]. As sequelae did not occur in any of the study groups, the authors concluded that initial oral antibiotic treatment was noninferior to initial intravenous antibiotics, followed by oral therapy.

Despite advances in medicine reducing mortality, the morbidity associated with BJIs is substantial [6]. For example, 9% of the patients have required admission to the Paediatric Intensive Care Unit (PICU) in one European cohort [6]. Adverse outcomes were observed in a considerable number of patients in European studies, though most were minor, and only a few required ongoing medical care [22]. Sequelae are a significant concern, with rates ranging from 1.1% to 7.9% even with appropriate diagnosis and treatment [7]. Long-term sequelae like growth disturbance, bone deformity, and chronic infection have been reported in 2–40% of patients [22]. Moreover, persistent complaints such as pain, functional differences, and scar paraesthesia were reported by 28% of patients in one study, with 5% requiring ongoing medical care, while objective sequelae like pain, limited range of motion, unilateral axis deformity, or asymmetric gait were documented in 12% of participants [22]. Based on the Paediatric Overall Performance Score, 10.5% of patients demonstrated mild overall disability, 0.8% moderate disability, and 0.2% severe overall disability [6]. Septic arthritis of the hip is particularly prone to long-term sequelae, with reported rates as high as 47% in affected newborns and 66% in other affected children in some regions [22]. Patients with a combination of OM and SA tend to show more mild disabilities at discharge [6]. In view of the evidence, current guidelines suggest that patients be followed for a minimum of one year by a specialist to assess long-term complications [9]. Nonetheless, the identification of children at higher risk for complications is crucial for effective management [8]. Key predictors of complicated outcomes include older age, female sex, joint involvement, higher CRP levels, persistent fever, and a greater number of surgical interventions [7,10].

Our study has several limitations. As this was a single-centre study, the conclusions drawn from our survey cannot be generalized to the whole country. All retrospective studies are subject to residual bias. Moreover, a discrepancy in the follow-up process was also attributable to the retrospective nature of the study and the fact that not all children were necessarily observed by the same orthopaedic specialist. Unfortunately, not all of them strictly adhered to the one-year follow-up recommended in the guideline of the European Society for Paediatric Infectious Diseases. Moreover, some cases with negative culturing results may represent either true-negative findings or culture-negative infections that could not be confirmed by the available diagnostic methods. Finally, the definition of complicated disease was arbitrary, which may have led to selection bias.

## 4. Materials and Methods

### 4.1. Study Design and Settings

A retrospective, single-centre observational cohort study was conducted between 2015 and 2022 at our tertiary referral centre, including children (aged ≤ 18 years) who were hospitalised with a diagnosis of AHO or SA within two months of symptom onset and without a history of penetrating wound infection or prior orthopaedic surgery. Eligible patients were identified through a systematic review of hospital admission logs, electronic medical records, and microbiology laboratory databases to ensure comprehensive case ascertainment. Children presenting with symptoms exceeding two months, as well as those with penetrating wound infection or surgical intervention of the affected area were excluded. The study was conducted in accordance with national ethical standards and the Declaration of Helsinki, and its protocol was approved by the Institutional Review Board of Heim Pál National Paediatric Institute (KUT-5/2023.01.23.).

### 4.2. Data Collection and Definitions

Data from the included patients were anonymously extracted to a standardized case report form. The data collected were as follows: (1) age, sex; (2) symptoms; (3) results of laboratory, microbiological, and radiological examinations; and (4) clinical outcomes and follow-up. AHO and SA diagnoses were prospectively ascertained. A detailed differential diagnostic algorithm of AHO and SA is given in Appendix A. Laboratory, microbiological, and radiological analyses were performed on all of the patients. All included patients were evaluated by a multidisciplinary team comprising of radiologists, infectious disease specialists, orthopaedic surgeons, and general paediatricians. Diagnostic and treatment decisions were made following team consensus, ensuring that each case underwent a structured assessment prior to inclusion in the study. AHO was defined as (1) fever, soft tissue swelling with warmth, or bone pain with limited joint mobility (minimum of 2), plus (2) radiological features of pathological bone resorption, subperiosteal/intraosseal abscess or necrosis, or bone sequestration (min. of 1 imaging, on ≥1 imaging modality, such as X-ray, ultrasonography or magnetic resonance imaging (MRI)), with or without (3) isolation of a causative pathogen from a blood or bone sample. SA was defined as (1) fever, joint swelling and warmth, joint pain and limited mobility (minimum of 2), plus (2) joint effusion demonstrated by ultrasonography, with or without (3) isolation of a causative pathogen from blood or joint fluid culture. A detailed table of characteristic findings of AHO and SA on imaging modalities is given in Appendix A. Complicated AHO was defined if intraosseous abscess or necrosis was proven by radiological or intraoperative visualization in our practice. Spondylodiscitis was defined as complicated AHO. AHO and SA were defined as acute if the symptom duration was ≤14 days and subacute if the symptom duration was between 14 days and two months at hospitalization. Microbiological diagnosis was performed by standard culturing methods from blood cultures and intraoperative surgical samples; PCR testing was not performed. Identification and in vitro antibiotic susceptibility testing of the isolated microorganisms were performed in accordance with the European Committee on Antimicrobial Susceptibility Testing (EUCAST) guidelines (https://www.eucast.org/clinical_breakpoints, accessed on 1 August 2025).

### 4.3. Clinical Outcomes and Follow-Up

The primary clinical outcome was the requirement for surgical intervention on day +30 post-diagnosis. The secondary clinical outcome was the occurrence of any long-term complication at the end of follow-up. A post-discharge follow-up was carried out by the paediatric orthopaedist involved in the in-hospital treatment. Any clinical symptom that impacts a child’s quality of life, such as persistent pain, limping, limb length discrepancy or stiffness, was defined as a long-term complication. All patients were followed retrospectively until the last documented hospitalization or outpatient visit.

### 4.4. Statistical Analysis

Continuous variables are reported as medians ± interquartile ranges (IQRs) or means ± standard deviations (SDs), depending on their distributions. Categorical variables are reported as absolute numbers (*n*) and percentages (%). Normality was checked with the Shapiro–Wilk test. For pairwise comparisons, the Mann–Whitney U test, χ^2^ test, or Fischer’s exact test was used. A two-tailed *p* value ≤ 0.05 was considered statistically significant. Calculations were performed via SPSS 23.0. The study adheres to the STROBE Statement (STrengthening the Reporting of OBservational studies in Epidemiology, www.strobe-statement.org).

## 5. Conclusions

This study provides comprehensive insights into paediatric BJI in Hungary, identifying *S. aureus* as the predominant pathogen. The observed high incidence of complicated AHO, frequent need for surgical interventions, extended durations of intravenous antibiotic therapy, and a notable proportion of reoperations, especially in cases associated with MSSA, reflect trends seen in other European settings. While our findings are rooted in the Hungarian healthcare context, the diagnostic and treatment challenges we encountered are relevant to paediatric BJIs globally.

Overall, our study might highlight the importance of early, accurate diagnosis and coordinated, multidisciplinary management to minimize long-term complications, such as growth disturbances, fractures, and joint dysfunction. These findings support the clinical adoption of optimized diagnostic and intervention strategies not only in Hungary but also in similar healthcare settings internationally. Moreover, continued refinement of therapeutic strategies, like shorter antibiotic courses and earlier oral switching, alongside a standardized post-discharge follow-up, is probably required to improve patient outcomes. Further prospective, multicentric research is needed to refine management protocols, ultimately improving outcomes for children with BJI worldwide.

## Figures and Tables

**Figure 1 antibiotics-14-00821-f001:**
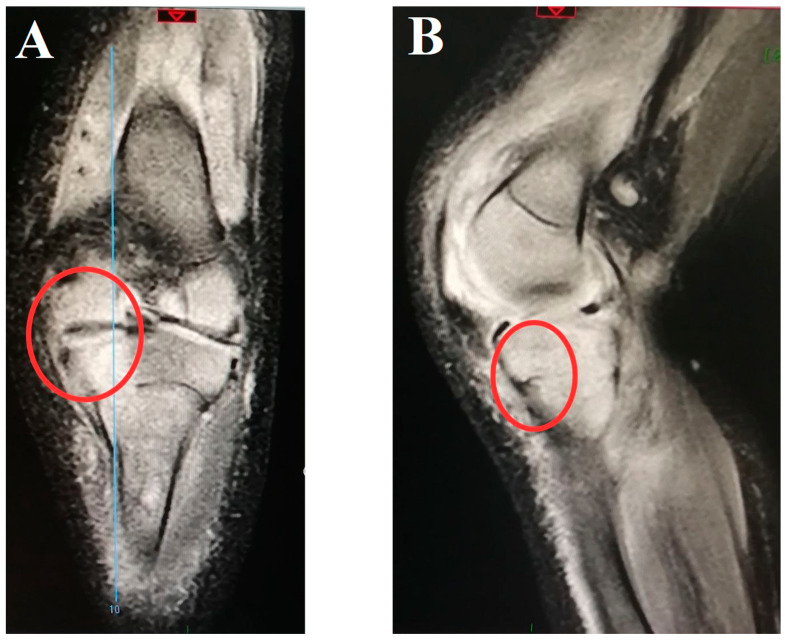
Magnetic resonance imaging of the right knee joint from the coronal (**A**) and sagittal (**B**) planes of a 4-month-old boy with infective arthritis of caused by methicillin-susceptible *Staphylococcus aureus* with consequent tibial osteomyelitis. Red circle: involvement of the growth zone.

**Figure 2 antibiotics-14-00821-f002:**
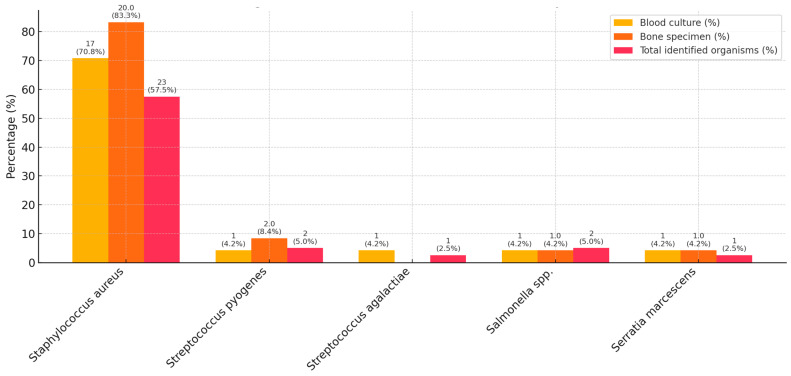
Causative organisms, isolated from clinical samples. Percentages are reported as the number of all isolates of the given bacterium per the number of patients in the cohort (*n* = 40), or all culture-positive cases of the given clinical sample. In the event of a pathogen being identified in both a blood culture and an intraoperative specimen, it was counted as one pathogen in the total cohort.

**Figure 3 antibiotics-14-00821-f003:**
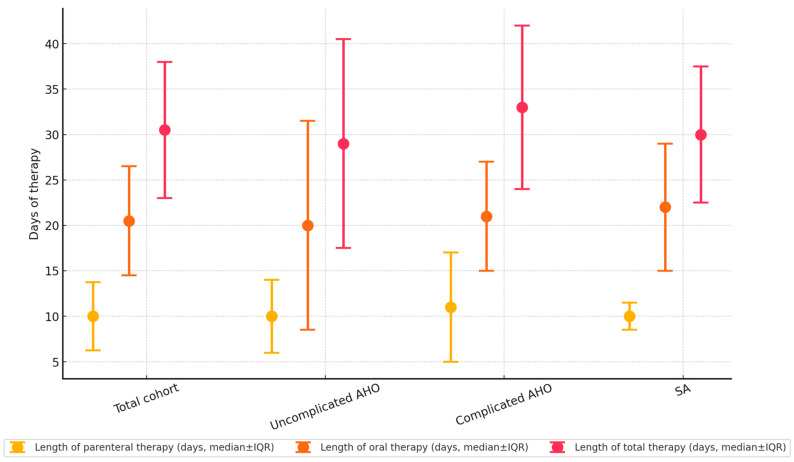
Lengths of parenteral, oral, and total antibiotic therapies (in days). Markers indicate median values, while whiskers represent the interquartile ranges.

**Table 1 antibiotics-14-00821-t001:** Baseline clinical and laboratory characteristics of included patients.

Parameter	Total Cohort (*n* = 40)	AHO (*n* = 30)	SA (*n* = 10)	*p* Value
Age (years, median ± IQR)	8.7 ± 9.9	9.4 ± 6.5	4.95 ± 8.1	0.16
Age cohorts (*n*, %) - <1 year - 1–5 years - 6–18 years	7 (17.5) 6 (15) 27 (67.5)	4 (13.3) 3 (10) 23 (76.6)	3 (30) 3 (30) 4 (40)	0.16
Male sex (*n*, %)	31 (77.5)	24 (80)	7 (70)	0.67
Infection type according to complications (*n*, %) - Uncomplicated - Complicated	18 (45.0) 22 (55.0)	8 (26.7) 22 (73.3)	10 (100.0) n.a.	<0.001 *
No. of patients receiving antibiotics prior to admission (*n*, %)	7 (17.5)	6 (20)	1 (10)	0.66
Length of symptoms prior to admission (days, median ± IQR)	4.0 ± 3.5	5.0 ± 5.0	2.0 ± 2.0	0.01 *
Infection type according to symptom length (*n*, %) - Acute - Subacute	36 (90) 4 (10)	26 (86.6) 4 (13.3)	10 (100) 0	0.56
Symptoms at admission (*n*, %) - Limb or joint pain - Fever - Joint movement restriction - Limping - Swelling of a limb or joint - Calor- - Pseudoparalysis - Erythema - Vomiting - Difficulty feeding - Irritability	39 (97.5) 37 (92.5) 29 (72.5) 25 (62.5) 22 (55) 12 (30) 10 (25) 9 (22.5) 7 (17.5) 5 (12.5) 5 (12.5)	29 (96.6) 27 (90) 20 (66.6) 20 (66.6) 17 (56.6) 10 (33.3) 4 (13.3) 7 (23.3) 6 (20) 5 (16.6) 3 (10)	10 (100) 10 (100) 9 (90) 5 (50) 5 (50) 2 (20) 6 (60) 2 (20) 1 (10) 0 (0) 2 (20)	1 0.56 0.23 0.46 0.73 0.69 0.007 * 1 0.66 0.31 0.58
Laboratory results at admission - White blood cell count (×10^9^/L) ^a^ - Absolute neutrophil cell count (×10^9^/L) ^b^ - Absolute lymphocyte count (×10^9^/L) ^b^ - Serum C-reactive protein (mg/L) ^b^ - Serum procalcitonin (ng/mL) ^a^ - Erythrocyte sedimentation rate (mm/h) ^b^	9.69 ± 6.74 8.59 ± 5.15 1.46 ± 0.96 121.7 ± 81.5 1.11 ± 4.63 44 ± 28	10.4 ± 6.54 8.44 ± 5.07 2.15 ± 1.87 130.6 ± 108.3 0.99 ± 5.5 43 ± 26	16.15 ± 10.2 9.49 ± 3.44 4.7 ± 4.65 85.6 ± 39.0 1.99 ± 0.5 49 ± 25	0.09 0.19 0.15 0.38 0.18 0.93

AHO: acute haematogenous osteomyelitis, SA: septic arthritis. ^a^ Values are reported as median ± IQR. ^b^ Values are reported as mean ± SD. * Statistically significant differences.

**Table 2 antibiotics-14-00821-t002:** Causative organisms with relevant in vitro antibiotic susceptibility data.

Antibiotic ^a^	*Staphylococcus aureus*	*Streptococcus pyogenes*	*Streptococcus agalactiae*	*Salmonella* spp.	*Serratia marcescens*
Penicillin	n.r.	2/2 (100)	n.r.	n.r.	n.r.
Ampicillin	n.r.	2/2 (100)	1/1 (100)	2/2 (100)	0/1 (0)
Amoxicillin/clavulanate	n.r.	n.r.	n.r.	n.r.	n.r.
Methicillin	22/23 (95.6)	n.r.	n.r.	n.r.	n.r.
Ceftriaxone	n.r.	2/2 (100)	1/1 (100)	2/2 (100)	1/1 (100)
Cefepime	n.r.	n.r.	n.r.	2/2 (100)	1/1 (100)
Meropenem	n.r.	n.r.	1/1 (100)	2/2 (100)	1/1 (100)
Erythromycin	20/23 (86.9)	2/2 (100)	1/1 (100)	n.r.	n.r.
Clindamycin	20/23 (86.9)	2/2 (100)	1/1 (100)	n.r.	n.r.
Tetracycline	22/23 (95.6)	2/2 (100)	1/1 (100)	n.r.	n.r.
*Trimethoprim*/*sulfamethoxazole*	23/23 (100)	n.r.	1/1 (100)	2/2 (100)	1/1 (100)
Vancomycin	1/1 (100)	n.r.	n.r.	n.r.	n.r.
Linezolid	n.r.	n.r.	n.r.	n.r.	n.r.
Moxifloxacin	23/23 (100)	n.r.	n.r.	n.r.	n.r.
Ciprofloxacin	n.r.	n.r.	n.r.	1/2 (50)	1/1 (100)
Gentamicin	23/23 (100)	n.r.	n.r.	n.r.	1/1 (100)

n.r.: not relevant/no data. ^a^ Ratios are reported as the number of all in vitro full sensitive isolates to the given antibiotic per the number of all isolates of the given bacterium from all culture-positive clinical samples.

**Table 3 antibiotics-14-00821-t003:** Characteristics of imaging studies of included patients.

Parameter	Total Cohort (*n* = 40)	AHO (*n* = 30)	SA (*n* = 10)	*p* Value
X-ray imaging performed ^a^ (*n*, %)	30 (75.0)	25 (83.3)	5 (50)	0.08
X-ray imaging positivity ^b^ (*n*, %)	9 (30)	7 (28)	2 (40)	1
X-ray imaging alterations ^c^ (*n*, %) - Bone resorption - Periosteal abnormality - Soft tissue abnormality	4 (40.0) 0 2 (20.0)	4 (57.1) 0 1 (14.2)	0 0 1 (10)	0.56 1 0.44
Ultrasonography performed ^a^ (*n*, %)	35 (87.5)	26 (86.6)	9 (90)	1
Ultrasonography positivity ^b^ (*n*, %)	30 (85.7)	22 (84.6)	8 (88.8)	1
Ultrasonography alterations ^c^ (*n*, %) - Intraarticular fluid - Soft tissue abnormality ^d^ - Myositis - Cellulitis - Periosteal abnormality - Deep venous thrombosis	21 (70) 12 (40) 4 (13.3) 3 (10) 3 (10) 1 (3.3)	13 (59.1) 11 (50) 3 (13.6) 2 (9.1) 3 (13.6) 1 (4.5)	8 (100) 1 (12.5) 1 (12.5) 1 (12.5) 0 0	1 0.23 1 1 0.56 1
Magnetic resonance imaging performed ^a^ (*n*, %)	32 (80)	28 (93.3)	4 (40)	0.001 *
Magnetic resonance imaging positivity ^b^ (*n*, %)	31 (96.8)	28 (100)	3 (75)	<0.001 *
Magnetic resonance imaging alterations ^c^ (*n*, %) - Soft tissue abnormality - Joint cavity breaching - Cellulitis - Subperiosteal abscess - Intraosseal abscess - Intraosseal necrosis - Deep venous thrombosis - Bone sequester	17 (54.8) 9 (29.0) 5 (16.1) 6 (19.3) 2 (6.4) 1 (3.2) 1 (3.2) 1 (3.2)	16 (57.1) 9 (32.1) 5 (17.8) 6 (21.4) 2 (7.1) 1 (3.5) 1 (3.5) 1 (3.5)	1 (33.3) - 0 0 0 0 0 0	0.02 * - 0.31 0.31 1 1 1 1

AHO: acute haematogenous osteomyelitis, SA: septic arthritis. ^a^ Ratios are given relative to the number of the total cohort (*n* = 40). ^b^ Ratios are given relative to the number of imaging studies performed. ^c^ Ratios are given relative to the number of positive imaging studies. ^d^ Not otherwise specified. * Statistically significant differences.

## Data Availability

The datasets generated and analysed during the current study are available from the corresponding author upon reasonable request.

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
