# Peer review of "Clinical Characteristics and Follow-Up of Children with Primary Haematogenous Osteomyelitis and Septic Arthritis: Eight Years of Experience from Hungary"

_antibiotics, 2025, doi:10.3390/antibiotics14080821_

Round 1
Reviewer 1 Report
Comments and Suggestions for Authors
The authors report on “Clinical characteristics and follow-up of children with primary haematogenous osteomyelitis and septic arthritis: eight years of experience from Hungary”.
The following comments and suggestions are provided to improve the manuscript and increase the number of potential readers.
- Introduction
Line 45. Although these infections are rare,
Suggested modification: Although these two clinical conditions are rare,
- MATHERIALS AND methods
Modify as: Materials and methods
- Data collection and definitions
Lines 65-73. AHO and SA diagnoses were prospectively ascertained. AHO was defined as (1) fever, soft tissue swelling with warmth, or bone pain with limited joint mobility (min-66 imum of 2), and (2) radiological features of pathological bone resorption, subperiosteal/intraosseal abscess or necrosis, or bone sequestration (min. of 1 imaging, on ≥1 imaging modality, such as X-ray, ultrasonography or magnetic resonance imaging [MRI]), with or without (3) isolation of a causative pathogen from a blood or bone sample. SA was defined as (1) fever, joint swelling and warmth, joint pain and limited mobility (minimum of 2), (2) joint effusion demonstrated by ultrasonography, with or without (3) isolation of a causative pathogen from blood or joint fluid culture.
Comment: For AHO, in case of negative culture from blood or joint fluid, it should be explained how it was considered the differential diagnosis with the conditions under the SAPHO syndrome umbrella and chronic non-bacterial osteitis (CNO). For SA in case of negative culture from blood or joint fluid, it should be explained how, especially in infants, it was considered the differential diagnosis of Aseptic synovitis of the hip, viral arthritis, Juvenile rheumatoid arthritis, Legg-Calve-Perthes disease, Lyme disease (borreliosis), Slipped capital femoral epiphysis, Reactive arthritis, Sickle cell anemia. It might be helpful and also interesting for the readers to include a diagnostic flow chart for the differential diagnosis and if any case of the above mentioned entities were observed.
- Clinical outcomes and follow-up
Lines 81-82. The primary clinical outcome was the requirement for surgical intervention on day +30 post-diagnosis.
Comment: The accuracy of the diagnosis of AHO and SA is of particular importance when surgical intervention is required in children and the issue should be addressed with a detailed description of the differential diagnosis to minimize the possibilities of errors.
- Results
3.1. Baseline characteristics
Lines 102-103. Within the AHO group, the majority of patients were classified as complicated (22/30, 102 73.3%),
Comment: The difference between complicated and non-complicated should be explained.
Lines 114-116. Among all children, a 114 causative microorganism was identified in 72.5% (29/40), with a higher rate in SA cases 115 (90%) than in complicated AHO (71.4%) and uncomplicated AHO (62.5%) cases.
Comment: The modality of identification of the causative microorganism should be specified with a comment regarding the negative cases (true negative/false negative).
3.3. Imaging characteristics
Lines 140-142. The overall positivity rates for X-ray radiography, ultrasonography and magnetic resonance imaging (MRI) were 33.3%, 85.7% and 96.9%, respectively.
Comment: The criteria and their significance for “positivity” for each modality should be specified to justify the significant difference between plain radiograph and US/MRI.
3.4. Antibiotic treatment strategies
Comment: If available, it would be interesting to know the microbiological characterization of various strains of the causative bacterial agent (single, multiple) and its influence on the choice of the therapeutic regimen.
- Discussion
Comment: The discussion might need modifications according to the suggestions and comments provided for the previous sessions.
- CONCLUSION
Please modifiy as 5. Conclusion
Comment: Although the study is conducted in Hungary, the findings can be of general interest and use in other countries too. Specific considerations for this cohort if present should be highlighted with suggested interventions to optimize the diagnostic tree and clinical outcome.
Author Response
Reviewer #4
Introduction
Line 45. Although these infections are rare, Suggested modification: Although these two clinical conditions are rare,
We appreciate the suggestion to clarify the wording. However, as another reviewer requested a complete rewrite of the Introduction, this section was also comprehensively revised.
MATHERIALS AND methods Modify as: Materials and methods
Corrected, as requested.
Data collection and definitions
Lines 65-73. AHO and SA diagnoses were prospectively ascertained. AHO was defined as (1) fever, soft tissue swelling with warmth, or bone pain with limited joint mobility (min-66 imum of 2), and (2) radiological features of pathological bone resorption, subperiosteal/intraosseal abscess or necrosis, or bone sequestration (min. of 1 imaging, on ≥1 imaging modality, such as X-ray, ultrasonography or magnetic resonance imaging [MRI]), with or without (3) isolation of a causative pathogen from a blood or bone sample. SA was defined as (1) fever, joint swelling and warmth, joint pain and limited mobility (minimum of 2), (2) joint effusion demonstrated by ultrasonography, with or without (3) isolation of a causative pathogen from blood or joint fluid culture. Comment: For AHO, in case of negative culture from blood or joint fluid, it should be explained how it was considered the differential diagnosis with the conditions under the SAPHO syndrome umbrella and chronic non-bacterial osteitis (CNO). For SA in case of negative culture from blood or joint fluid, it should be explained how, especially in infants, it was considered the differential diagnosis of Aseptic synovitis of the hip, viral arthritis, Juvenile rheumatoid arthritis, Legg-Calve-Perthes disease, Lyme disease (borreliosis), Slipped capital femoral epiphysis, Reactive arthritis, Sickle cell anemia. It might be helpful and also interesting for the readers to include a diagnostic flow chart for the differential diagnosis and if any case of the above mentioned entities were observed.
We agree with the reviewer, therefore we added a Supplementary file and details to the manuscript, detailing the steps of differential-diagnosis.
Clinical outcomes and follow-up
Lines 81-82. The primary clinical outcome was the requirement for surgical intervention on day +30 post-diagnosis. Comment: The accuracy of the diagnosis of AHO and SA is of particular importance when surgical intervention is required in children and the issue should be addressed with a detailed description of the differential diagnosis to minimize the possibilities of errors.
We agree with the reviewer, therefore we added a Supplementary file and details to the manuscript, detailing the steps of differential-diagnosis.
Results 3.1. Baseline characteristics Lines 102-103. Within the AHO group, the majority of patients were classified as complicated (22/30, 102 73.3%). Comment: The difference between complicated and non-complicated should be explained.
Thank you for this comment. We agree that the distinction between complicated and non-complicated AHO should be clear. This has already been addressed in the manuscript (original line 308), where we define complicated AHO as cases in which an intraosseous abscess or necrosis was confirmed by radiological or intraoperative visualization. Additionally, all cases of spondylodiscitis were classified as complicated AHO.
Lines 114-116. Among all children, a 114 causative microorganism was identified in 72.5% (29/40), with a higher rate in SA cases 115 (90%) than in complicated AHO (71.4%) and uncomplicated AHO (62.5%) cases. Comment: The modality of identification of the causative microorganism should be specified with a comment regarding the negative cases (true negative/false negative).
Thank you for raising this important point. The modality of pathogen identification was updated in the Methods section, where we state that microbiological diagnosis was performed using standard culturing methods from blood cultures and intraoperative surgical samples; PCR testing was not performed. Consequently, cases with negative results may represent either true-negative findings or culture-negative infections that could not be confirmed by the available diagnostic methods. This was added to the limitations.
3.3. Imaging characteristics
Lines 140-142. The overall positivity rates for X-ray radiography, ultrasonography and magnetic resonance imaging (MRI) were 33.3%, 85.7% and 96.9%, respectively. Comment: The criteria and their significance for “positivity” for each modality should be specified to justify the significant difference between plain radiograph and US/MRI.
We agree with the reviewer, therefore we added a Supplementary file and details to the manuscript, detailing the characteristic findings on imaging.
3.4. Antibiotic treatment strategies Comment: If available, it would be interesting to know the microbiological characterization of various strains of the causative bacterial agent (single, multiple) and its influence on the choice of the therapeutic regimen. Discussion Comment: The discussion might need modifications according to the suggestions and comments provided for the previous sessions.
Additional details regarding targeted antimicrobial de-escalation and escalation strategies, as well as the prevalence of polymicrobial infections, have been added to the revised manuscript.
CONCLUSION Please modifiy as 5. Conclusion. Comment: Although the study is conducted in Hungary, the findings can be of general interest and use in other countries too. Specific considerations for this cohort if present should be highlighted with suggested interventions to optimize the diagnostic tree and clinical outcome.
Thank you for this valuable comment. In response, we have added several new references and rephrased multiple sections of the Discussion to address the suggested points. The Conclusion has been modified accordingly to emphasize that, while this study was conducted in Hungary, the findings are probably applicable to other countries.
Reviewer 2 Report
Comments and Suggestions for Authors
The study offers valuable, particularly region-specific insights into pediatric bone and joint infections management.
Statistical analysis missing in Table 2 A and B.
It will be better-if you include few healthy or alternative-disease control group (e.g., trauma, non-infective arthritis), which limits the strength of conclusions regarding diagnostic specificity and treatment efficacy.
You may include some representative image,
Sample size is too less, some tables like Table 2A may be convert to bar diagram with error bar.
Author Response
Reviewer #1
The study offers valuable, particularly region-specific insights into pediatric bone and joint infections management.
Statistical analysis missing in Table 2 A and B.
We appreciate the reviewer’s assessment of our manuscript. We would like to clarify that statistical analysis was not performed for Table 2A and 2B on purpose, rather than being inadvertently omitted. This decision was made because no clinically meaningful subgroups could be defined within the data presented in Table 2A and 2B that would allow an interpretable statistical comparison. To avoid potential over-interpretation, we therefore opted to present these data only in a descriptive fashion.
It will be better-if you include few healthy or alternative-disease control group (e.g., trauma, non-infective arthritis), which limits the strength of conclusions regarding diagnostic specificity and treatment efficacy.
We thank the reviewer for this thoughtful suggestion. However, our study was not designed as a case-control investigation; rather, it was planned as an observational cohort focusing exclusively on pediatric patients with confirmed bone and joint infections. Consequently, data on healthy individuals or those with non-infectious conditions (eg. trauma, non-infective arthritides etc.) were not collected.
Furthermore, we think that healthy children cannot meaningfully serve as a comparator group for outcomes such as surgery rates, antibiotic duration or complication profiles, as these outcomes would be zero in an uninfected population. In addition, traumatic injuries and non-infectious arthritides have distinct pathophysiological mechanisms and clinical courses, making their inclusion as comparators less clinically relevant and potentially misleading when interpreting our findings.
For these reasons, the study was intentionally limited to infected patients only, to maintain clinical relevance and ensure the external validity of our conclusions.
You may include some representative image.
We thank the reviewer for this helpful suggestion. In response, we have added a representative figure to the manuscript.
Sample size is too less, some tables like Table 2A may be convert to bar diagram with error bar.
We appreciate the reviewer’s input regarding data presentation. Therefore, we generated two Figures from two previous Tables (Tables 2A and 4) for additional clarification.
Reviewer 3 Report
Comments and Suggestions for Authors
Dear Authors,
I was pleased to review the paper entitled " Clinical characteristics and follow-up of children with primary haematogenous osteomyelitis and septic arthritis: eight years of experience from Hungary" - MDPI – The present paper is very interesting, it focuses on a relevant clinical scenario, for orthopedics, potentially influencing the surgical and clinical practice for the management of haematogenous osteomyelitis. Below is my revision, highlighting some grammatical and conceptual points that need to be corrected or clarified.
- Title: The title gives a fine idea of the topic to be covered.
- The section heading "MATHERIALS AND methods" contains a typographical error and should be corrected to "MATERIALS AND METHODS".
- There is inconsistent use of British and American English. For instance, the word "paediatric" appears in the title, while "hematogenous" (American spelling) is used in the abstract. Ensure consistency throughout the manuscript according to the journal's style guide. The abbreviation BJIs is introduced without explanation in the abstract and again in the introduction. It should be defined as "bone and joint infections" upon first use.
PCT (procalcitonin) is first mentioned in the results section (line 107) but not defined until the abbreviations list. It should be defined upon first mention.
Statistical abbreviations such as IQR and SD are used in Table 1 without being defined in the table legend or main text. These should be introduced clearly.
The decimal formatting is inconsistent, mixing commas (e.g., 73,3%) and dots (e.g., 8.7 years). This should be standardized, preferably using the decimal point format in accordance with international scientific conventions.
- The follow-up duration is reported as a median of 115±436 days, which is highly inconsistent and suggests extreme variability. The authors should clarify the follow-up range and explain how this aligns with recommendations to follow patients for at least one year.
- The sentence stating that 75% of patients with uncomplicated osteomyelitis underwent MRI, which “we intend to discontinue in the future,” needs clarification. If this decision is based on guidelines, provide a specific reference. If not, explain the rationale.
- The very low MRSA prevalence (2.5%) in the cohort contrasts with national surveillance data reporting a rate of 11.1–21.5% in S. aureus infections. The authors should discuss whether this difference is due to age-specific trends, local epidemiology, or possible selection bias.
- The definition of “complicated AHO” appears operational but lacks a reference. If this classification is original, it should be clearly stated. Otherwise, cite validated criteria or similar studies.
- The authors cite national recommendations for MRSA treatment but state that empirical coverage is not considered standard for community-acquired BJIs. Given the impact of MRSA on clinical outcomes, a clearer justification of the local practice is warranted.
In the sentence “a first generation cephalosporin for 20–30 days,” the article “a” is missing before “first-generation”.
In Table 1, it would improve readability to mark statistically significant p-values with asterisks or other visual indicators.
- The decision to use follow-up MRI in uncomplicated osteomyelitis cases is not aligned with the literature cited in the discussion. The justification for this practice—and for its planned discontinuation—should be strengthened or removed.
- Discussion: Infection and septic arthritis is a terrible complication in orthopaedic surgery that can affect both paediatric and adult patients (you could add from doi: 10.3390/medicina58111537)
Overall, the discussion is thorough and well supported by the literature, but some conclusions would benefit from greater clarity and precision, particularly when deviating from standard clinical practices or when interpreting results with wide variability.
The paper generally is well written and needs only minor changes.
Author Response
Reviewer #2
Dear Authors, I was pleased to review the paper entitled "Clinical characteristics and follow-up of children with primary haematogenous osteomyelitis and septic arthritis: eight years of experience from Hungary" - MDPI – The present paper is very interesting, it focuses on a relevant clinical scenario, for orthopedics, potentially influencing the surgical and clinical practice for the management of haematogenous osteomyelitis. Below is my revision, highlighting some grammatical and conceptual points that need to be corrected or clarified.
Overall, the discussion is thorough and well supported by the literature, but some conclusions would benefit from greater clarity and precision, particularly when deviating from standard clinical practices or when interpreting results with wide variability. The paper generally is well written and needs only minor changes.
Title: The title gives a fine idea of the topic to be covered. The section heading "MATHERIALS AND methods" contains a typographical error and should be corrected to "MATERIALS AND METHODS".
Corrected, as suggested.
There is inconsistent use of British and American English. For instance, the word "paediatric" appears in the title, while "hematogenous" (American spelling) is used in the abstract. Ensure consistency throughout the manuscript according to the journal's style guide. The abbreviation BJIs is introduced without explanation in the abstract and again in the introduction. It should be defined as "bone and joint infections" upon first use.
We thank the reviewer for this observation. The manuscript has been revised to ensure consistent use of British English throughout (e.g., haematogenous instead of hematogenous). Additionally, the abbreviation BJI has been defined as bone and joint infections upon its first mention in the abstract and introduction.
PCT (procalcitonin) is first mentioned in the results section (line 107) but not defined until the abbreviations list. It should be defined upon first mention.
We appreciate the reviewer’s attention to detail. We would like to clarify that PCT is not mentioned in line 107 of the manuscript, instead its first mention occurs in line 125, where it is immediately defined as procalcitonin. The aabbreviations list was amended, as requested.
Statistical abbreviations such as IQR and SD are used in Table 1 without being defined in the table legend or main text. These should be introduced clearly.
Corrected, as suggested.
The decimal formatting is inconsistent, mixing commas (e.g., 73,3%) and dots (e.g., 8.7 years). This should be standardized, preferably using the decimal point format in accordance with international scientific conventions.
Corrected accordingly in Line 186, and in Tables 1 and 3.
The follow-up duration is reported as a median of 115±436 days, which is highly inconsistent and suggests extreme variability. The authors should clarify the follow-up range and explain how this aligns with recommendations to follow patients for at least one year.
We thank the reviewer for this important observation. In response, we have clarified the follow-up variability in the Discussion / limitations section of the manuscript. The revised text now states: “Moreover, a discrepancy in the follow-up process was also attributable to the retrospective nature of the study and the fact that not all children were necessarily observed by the same orthopaedic specialist. Unfortunately, not all patients strictly adhered to the one-year follow-up recommended in the guideline of the European Society for Paediatric Infectious Diseases.” We feel that this addition explains the inconsistency in follow-up duration and acknowledges deviation from the recommended one-year follow-up period.
The sentence stating that 75% of patients with uncomplicated osteomyelitis underwent MRI, which “we intend to discontinue in the future,” needs clarification. If this decision is based on guidelines, provide a specific reference. If not, explain the rationale.
We thank the reviewer for pointing out the need for clarification. We have revised the manuscript to explain the rationale and added a guideline reference. The updated text now states: “We do not intend to continue this practice in the future, as the Infectious Diseases Society of America guideline states that follow-up MRI is not always necessary in uncomplicated cases (3).” This addition might provide a clearer justification for the decision and cites the relevant guideline, as requested.
The very low MRSA prevalence (2.5%) in the cohort contrasts with national surveillance data reporting a rate of 11.1–21.5% in S. aureus infections. The authors should discuss whether this difference is due to age-specific trends, local epidemiology, or possible selection bias.
We thank the reviewer for this important comment. We have expanded the discussion to address this discrepancy. The revised text now states: “It is hypothesised that this discrepancy is age group–specific in comparison to the national population; therefore, it is conceivable that Hungary may exhibit a lower overall prevalence of MRSA among children. It is also important to note that the above national figure refers to both community-associated and healthcare-associated infections alike, while there were no hospital-acquired infections among our patients. In view of the low prevalence of MRSA in children, we do not consider empirical MRSA coverage necessary.” This addition clarifies the possible explanations for the difference in its broader epidemiological context.
The definition of “complicated AHO” appears operational but lacks a reference. If this classification is original, it should be clearly stated. Otherwise, cite validated criteria or similar studies.
We thank the reviewer for highlighting this point. We confirm that the definition of complicated AHO used in this study is our own operational definition. We have revised the manuscript to state explicitly that this classification was developed for the purposes of the present study.
The authors cite national recommendations for MRSA treatment but state that empirical coverage is not considered standard for community-acquired BJIs. Given the impact of MRSA on clinical outcomes, a clearer justification of the local practice is warranted.
Please refer to our answer above, concerning MRSA prevalence.
In the sentence “a first generation cephalosporin for 20–30 days,” the article “a” is missing before “first-generation”.
The “a” was added to the appropriate line.
In Table 1, it would improve readability to mark statistically significant p-values with asterisks or other visual indicators.
Corrected, as indicated.
The decision to use follow-up MRI in uncomplicated osteomyelitis cases is not aligned with the literature cited in the discussion. The justification for this practice—and for its planned discontinuation—should be strengthened or removed.
Please refer to our answer above, concerning the usage of MRI.
Discussion: Infection and septic arthritis is a terrible complication in orthopaedic surgery that can affect both paediatric and adult patients (you could add from doi: 10.3390/medicina58111537)
We thank the reviewer for the suggested reference. However, upon review, we found that the cited article does not strictly align with the primary aim and scope of our study („Continuous Cold Flow Device Following Total Knee Arthroplasty: Myths and Reality”). Therefore, we opted not to include it in the revised manuscript to maintain our original consistency.
Reviewer 4 Report
Comments and Suggestions for Authors
Dear authors, I consider the article to be quite interesting from a medical point of view, but in the current version there are several deficiencies that make it impossible for me to approve the article. Below I share my observations, which once addressed, will improve the quality of the article for its subsequent reconsideration for publication in the journal:
- Section "1. Introduction" is too brief. I recommend expanding its length to include important information for understanding the diseases mentioned, including etiology, symptoms, diagnosis, and other relevant characteristics. I also suggest providing an overview of the impact of these pathologies on human health and public health worldwide, in Europe, and in Hungary. This will improve the quality of this section and ensure readers understand the article more fully.
- In section "2.1. Study design and settings" I recommend that the inclusion and exclusion criteria taken into account in the patients be explained in more detail.
- In section "2.2. Data collection and definitions" it is mentioned that "The identification and in vitro antibiotic susceptibility testing of the microorganisms were performed in accordance with the guidelines of the European Committee on Antimicrobial Susceptibility Testing (EUCAST)", I suggest that these guidelines be briefly explained, it should also be mentioned that laboratory, microbiological and radiological analyses were performed on the patients, this will allow non-specialist readers to properly understand the methodologies used in the research.
- Section “2.4 Statistical Analysis” should be improved, taking into account that it should include a description of the data, explain in more detail the statistical analysis performed and the processing used in the research to ensure that the “Materials and Methods” section is as complete as possible.
- In section "3.1. Baseline characteristics," I suggest that the article text only include the percentage, not the number of people with the aforementioned characteristic. This lends more formality to the presentation of results. Furthermore, nothing is mentioned about laboratory results, so I recommend explaining them to improve the quality of this section.
- Similarly, in section "3.2. Microbiological characteristics" I recommend only showing the percentage and not the number of people with the characteristic. In addition, nothing is mentioned about the results of antimicrobial resistance. I consider it important to explain these results, but this will highlight the results obtained in the research.
- In the text of section "3.3. Imaging characteristics" not all the results of Table 3 are mentioned, I recommend that these results be explained, remember that explaining the results better allows to show their importance in the context of the research.
- In section "4. Discussion" I recommend that you do not add subsections, in general mention important information, but consider that the results obtained in your research are not adequately discussed and are not well contrasted with results obtained in research by other authors, I recommend that each and every one of the results obtained in your research are discussed more adequately with information from bibliographic sources both European and worldwide, this will allow to adequately highlight the importance of your results in human health and public health in your country, Europe and the world.
- Section "5. Conclusion" could be improved. I recommend briefly including the results of your research, its potential impact on human health, and whether it adequately meets the research objectives, to further emphasize your results and their impact on public health in Hungary, Europe, and the world.
Author Response
Reviewer #3
Dear authors, I consider the article to be quite interesting from a medical point of view, but in the current version there are several deficiencies that make it impossible for me to approve the article. Below I share my observations, which once addressed, will improve the quality of the article for its subsequent reconsideration for publication in the journal:
Section "1. Introduction" is too brief. I recommend expanding its length to include important information for understanding the diseases mentioned, including etiology, symptoms, diagnosis, and other relevant characteristics. I also suggest providing an overview of the impact of these pathologies on human health and public health worldwide, in Europe, and in Hungary. This will improve the quality of this section and ensure readers understand the article more fully.
We thank the reviewer for this valuable suggestion. In response, we substantially rewrote and expanded the entire Introduction section to provide a more comprehensive overview. The revised section now includes a description of the etiology, clinical presentation and diagnostic challenges of pediatric bone and joint infections, and an overview of the epidemiological impact of these infections worldwide, across Europe, and in Hungary. We feel that these revisions improve the overall background, and may ensure readers to be able to better understand the scope of our study.
In section "2.1. Study design and settings" I recommend that the inclusion and exclusion criteria taken into account in the patients be explained in more detail.
We thank the reviewer for this comment. We would like to clarify that no further inclusion or exclusion criteria were applied beyond what is already described. The study methodology has been fully documented in accordance with the STROBE statement to ensure reporting transparency. However, we provided additional details of patient recruitment.
In section "2.2. Data collection and definitions" it is mentioned that "The identification and in vitro antibiotic susceptibility testing of the microorganisms were performed in accordance with the guidelines of the European Committee on Antimicrobial Susceptibility Testing (EUCAST)", I suggest that these guidelines be briefly explained, it should also be mentioned that laboratory, microbiological and radiological analyses were performed on the patients, this will allow non-specialist readers to properly understand the methodologies used in the research.
We thank the reviewer for this suggestion. We would like to clarify that EUCAST is not a guideline itself, but rather an European scientific organisation dedicated to the standardisation and harmonisation of the in vitro antimicrobial susceptibility testings performed in clinical microbiology across Europe. Its methodology is widely recognised by specialists in infectious diseases and microbiology, and possibly the readership of Antibiotics, and is the universally accepted standard in European clinical laboratories. Therefore, due to space and scope constraints, we are unable to provide a detailed description of the full EUCAST workflow. However, to enhance clarity for non-specialist readers, we have included the EUCAST web address and specified that all patients underwent laboratory, microbiological, and radiological evaluations as part of their diagnostic work-up.
Section “2.4 Statistical Analysis” should be improved, taking into account that it should include a description of the data, explain in more detail the statistical analysis performed and the processing used in the research to ensure that the “Materials and Methods” section is as complete as possible.
We thank the reviewer for this comment. We would like to clarify that the study relied exclusively on simple descriptive statistical methods, such as medians, interquartile ranges and proportions, as the dataset and research questions did not require inferential analyses. The description provided reflects the complete methodology, and is consistent with the elements of the STROBE statement. Therefore, we feel that no additional detail is warranted beyond what has been reported.
In section "3.1. Baseline characteristics," I suggest that the article text only include the percentage, not the number of people with the aforementioned characteristic. This lends more formality to the presentation of results. Furthermore, nothing is mentioned about laboratory results, so I recommend explaining them to improve the quality of this section.
We appreciate the reviewer’s suggestion. However, we opted to report both the absolute numbers and percentages, in line with STROBE guidelines, to allow readers to fully understand the actual study subgroups in question. Regarding laboratory findings, details are already comprehensively presented in Table 1 – this approach was seeked to avoid redundancy in the main text, while ensuring data accessibily for readers. However, an additional sentence was added to the paragraph about laboratory results for clarification.
Similarly, in section "3.2. Microbiological characteristics" I recommend only showing the percentage and not the number of people with the characteristic. In addition, nothing is mentioned about the results of antimicrobial resistance. I consider it important to explain these results, but this will highlight the results obtained in the research.
Please refer to our previous answer regarding data representation in the manuscript. Although in vitro antibiotic susceptibility profiles are reported extensively in Table 2, an additional sentence was added to this paragraph for further clarification.
In the text of section "3.3. Imaging characteristics" not all the results of Table 3 are mentioned, I recommend that these results be explained, remember that explaining the results better allows to show their importance in the context of the research.
Some additional sentences were written in the paragraph, as requested.
In section "4. Discussion" I recommend that you do not add subsections, in general mention important information, but consider that the results obtained in your research are not adequately discussed and are not well contrasted with results obtained in research by other authors, I recommend that each and every one of the results obtained in your research are discussed more adequately with information from bibliographic sources both European and worldwide, this will allow to adequately highlight the importance of your results in human health and public health in your country, Europe and the world.
We thank the reviewer for these valuable comments. In response, we have removed the subsections from the Discussion section. Furthermore, we have expanded the discussion, ensuring that each of our findings is contrasted with relevant European and international literature.
Section "5. Conclusion" could be improved. I recommend briefly including the results of your research, its potential impact on human health, and whether it adequately meets the research objectives, to further emphasize your results and their impact on public health in Hungary, Europe, and the world.
We thank the reviewer for this thoughtful recommendation. In response, we have expanded the Conclusion section to briefly summarise the main results of our research and highlight the implications for public health in Hungary and beyond. However, given that the manuscript is already quite lengthy, we feel that the level of detail added is sufficient without making the section disproportionately long.
Round 2
Reviewer 1 Report
Comments and Suggestions for Authors
The comments and suggestions have been to some extent addressed; however, no information is provided on the number of patients with initial differential dx which resulted affected by entities other than AHO and SA. It looks improbable to this reviewer that the diagnostic tree was successful in 100% of the cases. Moreover, histopatological dx of SA in the pediatric population would be difficult to achieve on a synovial sample. It is important for the readers to know how the complex cases were not theoretically, but actually managed by the multidisciplinary team. It seems that there is a rush to publish this manuscript, whereas careful consideration should be given to review all data of the differential dx and their results. There are minor mistakes in the text and figure legends which should be edited before the final version of the manuscript is released.
Author Response
We thank the reviewer for the re-assessment of our manuscript and the additional comments provided. We believe these clarifications address the concerns raised, while keeping the focus on the intended scope of the manuscript.
We respectfully address the points raised as follows:
„The comments and suggestions have been to some extent addressed; however, no information is provided on the number of patients with initial differential dx which resulted affected by entities other than AHO and SA”
As our study was conducted in a tertiary referral hospital, all cases naturally underwent a structured differential diagnostic process real-time, as part of routine clinical care. This is standard procedure for any hospitalized paediatric patient, and is neither of scientific novelty, nor specific to the context of paediatric bone and joint infections.
We would also like to confirm once again that we only included patients whose final diagnosis was paediatric bone and joint infection; no comparison was sought with patients who had alternative diagnoses.
„It looks improbable to this reviewer that the diagnostic tree was successful in 100% of the cases.”
The diagnostic tree was incorporated into the manuscript following the explicit request of the Reviewer during an earlier round of comments. It was only provided to illustrate the general approach, not to imply that such algorithms are infallible or universally applicable in every clinical scenario. In real-world clinical practice, we think that diagnostic trees are useful only to a certain extent, which is precisely why expert consensus and multidisciplinary discussion play such a key role in final decision-making.
We would like to emphasize that the primary aim of this study was not to analyse the differential diagnostic pitfalls of paediatric bone and joint infections or to provide a diagnostic tree for attending physicians; the development of such a diagnostic algorithm lies beyond the scope of this study.
We respectfully prefer not to expand further on this topic beyond the level appropriate for such a retrospective observational study. All information on this topic that extends beyond the scope of our manuscript is available through the cited references and well‑respected handbooks of pediatric infectious diseases.
„Moreover, histopatological dx of SA in the pediatric population would be difficult to achieve on a synovial sample.”
We respectfully clarify that histopathology is naturally only performed when a pathological specimen is obtainable for collection during surgery (e.g. synovial membrane or tissue). Histopathological evaluation of septic arthritis in the paediatric population has its origins in the late 1970s (PMID: 524924). In cases where only synovial fluid is aspirated, it is sent for microbiological and laboratory fluid analysis, rather than histopathological analysis. This distinction is clearly outlined in our Supplementary file, for the sake of transparency.
„It is important for the readers to know how the complex cases were not theoretically, but actually managed by the multidisciplinary team.”
We agree that multidisciplinary management is important, and our institution consistently relies on such collaboration. This is precisely why we included the characteristics and outcomes of imaging studies, microbiological findings, surgical interventions (both short- and long-term), and conservative treatment strategies into our manuscript; to emphasize that each of these elements contributes meaningfully to everyday clinical practice.
However, this manuscript is not intended to serve as a protocol or comprehensive review on the operational dynamics of multidisciplinary team management. Readers with expertise in pediatric bone and joint infections are already familiar with these principles.
Nevertheless, we are willing to add a brief clarifying sentence to the Methods section, such as: “All included patients were evaluated by a multidisciplinary team comprising of radiologists, infectious disease specialists, orthopedic surgeons, and general pediatricians. Diagnostic and treatment decisions were made following team consensus, ensuring that each case underwent a structured assessment prior to inclusion in the study.”
„It seems that there is a rush to publish this manuscript, whereas careful consideration should be given to review all data of the differential dx and their results.”
We respectfully but firmly reject the implication that there is a “rush to publish”. This study represents the experience of eight years, while data collection and analysis took us more than a year to finish. Every patient was initially assessed by attending physicians in real time, and subsequently re-assessed for study eligibility for inclusion by the investigators. At each step, all available data were reviewed to ensure inclusion only of paediatric bone and joint infection cases, the best of our knowledge, as clearly described in our Methods section and the supplements.
Given the methodology of the manuscript and the highest level of reliability achievable in a retrospective observational study, we are uncertain as to what additional information the reviewer finds lacking at this stage.
We would like to assure the reviewer that we have applied rigorous methodology and case selection throughout this work, as described in full detail. The suggestion that our clinical case ascertainment was inadequate or that we lack the ability to evaluate our methodology is unfounded, and we would respectfully like to decline this insinuation.
Reviewer 4 Report
Comments and Suggestions for Authors
Dear authors, although I have identified improvements in certain sections of the article, other sections do not have significant modifications that would improve the quality of the article. Below I share some observations that will improve the quality of the article for its subsequent publication in the journal:
- Although the results of the laboratory analyses are presented in Table 1, the most important findings should be explained in the text, so that non-specialist readers can identify these results and subsequently fully understand their importance in the context of the research. Therefore, I again suggest that they be described in the text of the corresponding section "Baseline characteristics".
- In the "Microbiological characteristics" section, I understand that the results of antimicrobial resistance are shown in Table 2, but in the text of the section, I again suggest explaining the most important results, because this will allow readers to understand their impact on the research.
- The "Imaging characteristics" section has the same shortcomings as the previous sections. It is not enough to simply present the results in a table; they must be adequately explained in the text so that readers understand their impact in the context of the research.
- There are certain subsections of "Materials and Methods" that can be improved according to the observations of the previous review.
- The conclusion does not meet what I shared with you previously in the observations from the previous review; I consider it to be incomplete and need to be improved.
Author Response
Dear authors, although I have identified improvements in certain sections of the article, other sections do not have significant modifications that would improve the quality of the article. Below I share some observations that will improve the quality of the article for its subsequent publication in the journal:
Although the results of the laboratory analyses are presented in Table 1, the most important findings should be explained in the text, so that non-specialist readers can identify these results and subsequently fully understand their importance in the context of the research. Therefore, I again suggest that they be described in the text of the corresponding section "Baseline characteristics".
In the "Microbiological characteristics" section, I understand that the results of antimicrobial resistance are shown in Table 2, but in the text of the section, I again suggest explaining the most important results, because this will allow readers to understand their impact on the research.
The "Imaging characteristics" section has the same shortcomings as the previous sections. It is not enough to simply present the results in a table; they must be adequately explained in the text so that readers understand their impact in the context of the research.
We thank the reviewer for these constructive suggestions regarding the presentation of key findings. In response, we have revised the manuscript to provide more detailed explanations within the text for the sections on „Baseline Characteristics”, „Microbiological Characteristics” and „Imaging Characteristics”.
While the results remain summarized in Tables 1 and 2, we have now highlighted and discussed the most important findings in the corresponding text sections. We believe these additions will make the manuscript more accessible to non-specialist readers, and hopefully clarify the relevance of these results within the context of our study.
There are certain subsections of "Materials and Methods" that can be improved according to the observations of the previous review.
We appreciate the reviewer’s observation that certain subsections of the „Materials and Methods” could potentially be improved, and therefore, some further extensions were made. However, no further specific suggestions were provided in the previous review as to what additional content should be included exactly, and we think that a full adherence to the STROBE Statement was achieved within that paragraph.
Therefore, in the absence of concrete recommendations, we believe that the current level of detail in the statistical methods section is appropriate for the scope and design of this study, and we do not plan to expand this section any further.
The conclusion does not meet what I shared with you previously in the observations from the previous review; I consider it to be incomplete and need to be improved.
We thank the reviewer for their comment regarding the Conclusion section. In response to earlier feedback, we substantially rewrote this section; however, we feel that in doing so, the revised text became more of a condensed summary of the Discussion rather than a succinct conclusion.
Round 3
Reviewer 4 Report
Comments and Suggestions for Authors
Dear authors, I believe the quality of this article has improved substantially, meeting the requirements for publication in this journal. Best regards.